# Assessing the Design of Interactive Radial Data Visualizations for Mobile Devices

**DOI:** 10.3390/jimaging9050100

**Published:** 2023-05-14

**Authors:** Ana Svalina, Jesenka Pibernik, Jurica Dolić, Lidija Mandić

**Affiliations:** Faculty of Graphic Arts, University of Zagreb, 10000 Zagreb, Croatia; asvalina@grf.hr (A.S.); jurica.dolic@grf.unizg.hr (J.D.); lidija.mandic@grf.unizg.hr (L.M.)

**Keywords:** data visualization, design, graphical user interface, interaction, mobile visualization, perception

## Abstract

The growing use of mobile devices in daily life has led to an increased demand for the display of large amounts of data. In response, radial visualizations have emerged as a popular type of visualization in mobile applications due to their visual appeal. However, previous research has highlighted issues with these visualizations, namely misinterpretation due to their column length and angles. This study aims to provide guidelines for designing interactive visualizations on mobile devices and new evaluation methods based on the results of an empirical study. The perception of four types of circular visualizations on mobile devices was assessed through user interaction. All four types of circular visualizations were found to be suitable for use within mobile activity tracking applications, with no statistically significant difference in responses by type of visualization or interaction. However, distinguishing characteristics of each visualization type were revealed depending on the category that is in focus (memorability, readability, understanding, enjoyment, and engagement). The research outcomes provide guidelines for designing interactive radial visualizations on mobile devices, enhance the user experience, and introduce new evaluation methods. The study’s results have significant implications for the design of visualizations on mobile devices, particularly in activity tracking applications.

## 1. Introduction

In today’s fast-paced world, the use of mobile devices and mobile applications is on the rise. These applications present users with visualizations of significant data sets. They combine text and graphics with data visualizations to enable users to quickly recognize patterns, provoke insights, and encourage exploration. Information visualizations are common within health, fitness, and physical activity monitoring applications [1]. They enable the user to get an immediate understanding of personal fitness trends and how they change with exercise. The motivational impact of visualizations as a form of feedback has been recognized by the mobile application industry. According to Sydow [2], there was a 30% increase in global downloads of health and fitness applications in 2020 compared to the previous year. Furthermore, there was a surge in the supply of such applications with the launch of over 71,000 health and fitness applications worldwide in 2020, a 13% increase from the previous year. With the rise in the number of applications, there is an increasing need for design guidelines, especially for radial visualizations, which are a popular and frequently used form of information visualization in applications related to physical activity monitoring [3]. Physical activity monitoring involves the collection of data through sensors installed in mobile or wearable devices like smartwatches. The collected data is then displayed on an interface, enabling users to track their activity levels and take appropriate measures to either increase or decrease their activity as needed. However, the main challenge for effective mobile data visualization and interaction design is the small screen size. When it comes to data display, there is always a need for a compromise between the amount of data and its presentation. Keeping this in mind, the visualizations typically used are designed for quick glances. This study aims to provide guidelines for designing interactive visualizations on mobile devices and new evaluation methods based on the results of an assessment of interactive radial visualizations. The study’s contributions can be summarized as follows: (1) extraction of four radial visualization types common for fitness apps; (2) guidelines for designers; and (3) guidelines for conducting similar research studies. The paper is structured as follows: Section 1 presents the introduction and related works; Section 2 outlines the materials and methods employed in our user study and our research methods; Section 3 reports our experimental findings; Section 4 presents the discussion; and Section 5 synthesizes our results, highlights our contributions and limitations, and proposes future research directions.

When talking about visualization types, Burch and Weiskopf [4] have categorized shapes used to map visual data into two types: Cartesian and radial. Cartesian visualizations encode data within the Cartesian coordinate system, while radial visualizations encode data within the polar coordinate system [5]. Radial visualizations take on circular shapes such as circles, ellipses, spirals, and rings [3,4,5,6]. They are transformed representations of linear visualizations and require clear arguments for their use. Although they are often considered more visually appealing, spatially efficient, and natural compared to linear visualizations, they require greater cognitive effort for users to understand the data [4,7]. Some examples of radial visualizations include pie charts, Nightingale graphs (rose diagrams/polar area charts), star plots (radar plots/spider plots), donut graphs, and others [3,4]. Radial visualizations present complex graphic structures such as sectors, segments, and rings, resulting in asymmetry because elements closer to the center have less space than external ones, leading to distortion within the visualization [4].

Numerous studies have been conducted on radial visualizations, covering various topics such as interaction [8,9], reviewing radial methods for information visualization [3], and examining different types of radial visualizations on smartwatches [10]. Some authors have proposed new types of radial visualizations, including hypergraphs [11] and intercept graphs [12], while other studies have compared radial and linear graphs [5,6,7,13,14,15]. However, despite the emergence of several radial visualization techniques since 2010 [5], only a few research papers provide design guidelines. Moreover, there is a lack of consolidated evaluation of mobile interactions for visualizations that require one-dimensional selection [9].

Despite past research and literature indicating that radial visualizations may lead to misinterpretation by users due to column length and angle issues, they remain widely used in mobile applications. Radial visualizations have been defended for their elegant form, symmetry, completeness, and closure [4], but these should not be the sole justifications for their use. In fact, effective visualizations are characterized by several other factors, as identified by Cairo [16]. These characteristics are truthfulness, functionality, beauty, insight, and enlightenment. Additionally, when designing visualizations, it is imperative to consider the model of information processing in human visual perception as proposed by Ware [17]. This model highlights the parallel processing involved in extracting low-level features of the visual scene, pattern perception, and visual cognition.

The role of interactivity in information visualizations is crucial since it enables the transformation of data and the display of a comprehensive view. By engaging with interactive features, users can actively participate in creating fresh insights and drawing conclusions from the data presented. For mobile visualizations, an effective interaction technique is paramount, as it can determine the usability of the application and support the user’s goals. Conversely, a frustrating interaction technique can lead to poor usability and hinder the visualization’s effectiveness [9]. Our objective is to enrich the collection of best practices and advance knowledge regarding the subjective evaluation of interactive radial information visualizations on mobile devices based on our observations of user behavior and patterns. To delve deeper into this topic, we undertook a research study to assess users’ subjective evaluation of the design of interactive radial visualizations displaying fitness-related data on mobile devices.

## 2. Materials and Methods

The primary aim of our paper was to investigate how users perceived four different types of interactive radial visualizations, with a focus on usability metrics such as memorability, readability, understanding, enjoyment, engagement, and overall impression. Through our findings, we seek to establish design guidelines for radial visualizations used in physical activity monitoring applications. The study consisted of two parts: a comparative analysis and an online survey. In this chapter, we provide a detailed description of the experimental setup used in our study as well as the methods employed. We outline the procedures and tools we used to ensure the rigor and validity of our investigation.

### 2.1. Stimuli and Apparatus

The initial stage of our research involved conducting a comparative analysis of 16 mobile applications within the health and fitness category. To be included in this review, apps had to meet certain inclusion criteria: (1) be in the health and fitness category; (2) the purpose of the app involve tracking activity through sensors; and (3) use data visualizations. We searched the apps on the Google Play Store with the following keywords: fitness, tracking, track, fit, and health. We identified sixteen apps that fulfilled all these criteria (Table 1). Our focus was on the characteristics of the radial information visualizations used within these applications. Instead of examining the applications in their entirety, we solely concentrated on the sections where information visualizations were utilized. Our objective was to determine the types of visualizations, data, and colors used, as well as the data sets. This information served as a foundation for designing radial visualizations (test samples) for the main part of our research.

What we concluded from the comparative analysis is that there are four types of circular visualizations that are used, namely radial bar charts, polar area charts, half-radial bar charts, and donut charts (Table 1). Of these, the radial bar chart is the most commonly used, followed by the polar area chart and the half-radial and donut charts. Interestingly, radial bar charts are not always used in combination, but in some cases, individual displays are also used (where only one set of data is displayed on the visualization). When we look at the number of data sets, data in three and four sets are most commonly used, followed by one set and, to a lesser extent, two, five, and six data sets. When comparing colors, the most frequently used color is green, followed by blue shade 1, orange, red, and purple, while blue shade 2, pink, and yellow are used less frequently.

Based on the comparative analysis insights, we designed a high-fidelity prototype [18,19,20,21] of an imaginary mobile fitness application called “TRACKIT”. “TRACKIT” is a mobile app that enables users to track their daily physical activity. The main element on the home screen is a radial data visualization that displays four categories of data: steps, calories, activity, and sleep. Through visualization and the use of different colors, users can quickly see how close they are to reaching their goals without needing to read the exact numerical data. In addition to visualization, users can gain insight into their activity for certain time periods by manipulating the interactive timeline element that shows data for a one-day range. Below the visualization, users can find accurate numerical data for each category, along with the goal they have set for themselves. The app uses quantitative, continuous numerical data.

The four visualizations are based on the same set of data: radial bar chart (VIZA), polar area chart (VIZB), half-radial bar chart (VIZC), and donut chart (VIZD) (Figure 1). The function of all four visualizations is comparison (showing the differences/similarities between values) and proportions (using size/area to show differences/similarities between values or parts of a whole). The radial bar chart is a columnar graph transformed into a polar coordinate system. In the polar area chart (Nightingale rose chart), each category or data interval is divided into equal segments on the graph, and the distance of the segment from the center of the polar axis depends on the value it represents. The half-radial bar chart is a radial bar chart in which the full circle is not 360 degrees but 180 degrees. The donut chart is a pie chart that has a part removed from the center and is modified in such a way that it is divided into equal segments [22,23].

The independent variables were the type of radial visualization (Figure 1) and the type of interaction (drag and touch). The dependent variables were accuracy, subjective rating, task completion time, and number of taps. We used the same data type, data set (four categories), mobile application type, and visual encoding for all test samples. The drag interaction was used on VIZA and VIZC, and the touch interaction (tap-based) on VIZB and VIZD. When interacting by touch, the user selects the object with the touch of a finger. This technique is direct, simple, and fast, but only provides visual feedback after the interaction. On the other hand, when interacting by dragging, the user pulls the indicator and thus sees the data change in real-time. In this way, the interaction provides continuous feedback on the position of the data [9]. We also described the “TRACKIT” application usage scenario to the users as follows: “Imagine it is currently 6 p.m., and you have opened your activity tracking application. You are interested in how many activities you did at noon. Drag the indicator to noon (12 o’clock) and check the fulfillment of the goals for all four activities (steps, calories, activity, and sleep) in the shortest possible time. Once you’re finished, click the “Finish Task” button.” We conducted the test through the online tool Useberry, version 2.1 (Useberry User Testing Technologies P.C. (“Useberry”), Athens, Greece) [24], within which we set up a prototype of a mobile application designed in Figma tool. The survey was open between 8 July and 24 July 2022, with a total of 32 respondents.

### 2.2. Methods

In the initial phase, we utilized comparative analysis, while in the experimental phase, we relied on a subjective evaluation approach using survey questionnaires and assessment scales to assess participants’ interaction with visualizations. The literature points out that mobile applications cannot be tested well enough in a typical laboratory setting because environmental and context conditions are essential for use (subject movement, weather conditions, other people around subjects, etc.) and cannot be simulated in the laboratory [25]. Therefore, we decided to conduct an unmoderated study to create as realistic conditions as possible for the use of the mobile application. In this way, users used the application in their natural environment and on their mobile devices, which could vary in size of display. In addition, mobile devices are most often used in scenarios with fragmented attention, and for this reason, they should also be tested in such settings [25].

Our study aimed to investigate various aspects such as memorability, readability, understanding, enjoyment, and engagement. To evaluate the memorability of each graph type, participants were required to recall the value after interacting with the prototype. For readability, they needed to read the exact data, and for understanding, they had to compare the values of different categories. Interaction was examined through the categories of engagement and subjective assessment, while enjoyment (aesthetics) was evaluated based on the attractiveness of each display and subjective assessment. Based on the aforementioned, we formulated the following research questions:Q1/Layout: Is there a statistically significant difference between the four types of radial visualizations in the subjective evaluation of the data presented in the visualization?Q2/Enjoyment: Which of the four types of radial visualizations is visually most appealing to observers?Q3/Understanding: Which of the four types of visualizations is most understandable to observers?Q4/Interaction: Which of the two types of interaction (drag and touch) is more suitable for radial visualizations?

To avoid bias, we generated three surveys that varied solely in the sequence of visualizations presented. Subsequently, we merged all the outcomes into a single survey. The visualizations employed in the study facilitated three levels of perception and interpretation, namely value search, value comparison, and the ability to discern higher-level patterns such as trends and distribution [6]. Consequently, we have linked the survey questions to the levels and objectives. The questions addressed various aspects such as identification of maximum and minimum values, comparison of data, reading and interpretation of trends, and subjective assessment of the visualizations. To analyze the results, we employed several metrics, including accuracy, subjective responses, task execution time, and the number of taps. To facilitate the analysis process, we encoded the survey questions (Table 2).

## 3. Results

The collected data was quantitatively analyzed in the SPSS statistical program and Microsoft Excel. On average, it took respondents 17 min to complete the survey. Out of the 32 (N = 32) respondents, 14 (44%) were male and 18 (56%) were female. 21 (66%) were participants in the 25–34 age group, 5 (16%) in the 18–24 age group, and 3 (9%) of respondents in the 35–54 and 55+ age groups. The completed level of education for 16 (50%) respondents is graduate study; for 10 (31%) respondents, undergraduate study; for 4 (13%) respondents, high school; and for 2 (6%) respondents, a doctorate. When asked, “Do you use mobile applications to track activity (number of steps, calorie consumption, activities, etc.)?” 11 (34%) replied “Yes”, 8 (25%) “No”, and 13 (41%) replied “Sometimes”. Respondents who reported using mobile applications to track their physical activity provided an average rating of 3.21 for the usefulness of such apps. Respondents stated that they most often use Samsung Health (11), Google Fit (3), Apple Activity iOS (2), Garmin Connect (2), and OnePlus Health (1). Under “Other”, respondents also stated Huawei Health (3), Amazfit/Zepp (2), Nike Running App (1), and MyFitnessPal (1). Respondents also rated their competency to read and understand graphs and graphical representations, and the arithmetic mean of the average rating was 3.72. This suggests that most participants possess the ability to comprehend and interpret graphs effectively.

In the second part of the study, the participants performed their main tasks and answered questions. Observing the standard deviation values, we can conclude that the responses are widely dispersed (SD > 1.00) for almost all visualizations and questions (Table 3). Responses are not dispersed only for U2 (VIZB, VIZC, and VIZD) and U4 (VIZA, VIZC). Additionally, for VIZB, for almost all questions, dispersal is highest compared to VIZA, VIZC, and VIZD. Upon comparing the categories of memorability, readability, and understanding, it is evident that the lowest percentage of correct responses was in the memorability category. This outcome was anticipated as the memorability-related questions were relatively more demanding. On average, the participants’ interaction with the visualizations lasted approximately 30 s. VIZA stands out in terms of readability, with the highest percentage of correct responses (91% and 94%).

When it comes to understanding, it’s noteworthy that while participants had the lowest rate of correct answers (50% and 38%) when assessing the degree of understanding of the visualization, VIZC was rated as very understandable (AS = 4.19), which is also the highest rating among all visualizations. In terms of enjoyment, VIZB received the lowest average score in all answers, while VIZC had the highest score in four out of five questions. The engagement category was related to the interaction variable, and for this category, we compared A and C (DRAG-1) with B and D (TOUCH-2) visualizations. The average time participants spent interacting with each visualization was expressed using the geometric mean, which is more appropriate for time on task than the arithmetic mean. Participants spent the most time interacting with VIZB, which had touch interaction. The average number of taps was 6.6 for VIZA, 9.5 for VIZB, 7.3 for VIZC, and 9.6 for VIZD. Interestingly, visualizations with the same type of interaction (VIZA and VIZC-drag, VIZB and VIZD-touch) required the same number of taps. Based on this, we can conclude that drag interaction requires fewer taps. To obtain information on the precision of estimating the true population parameter, we also provide 95% confidence intervals (Cis) that indicate the plausible range of values for the population mean.

To determine if there is a statistically significant difference in responses, we conducted a nonparametric Kruskal–Wallis test. It showed that there was a statistically significant difference in responses for U3 (H = 10.910, *p* = 0.012) and TA (H = 9.497, *p* = 0.023) (Table 4). Based on the analysis of these two categories alone, we cannot conclude that there is a statistically significant difference in the evaluation of visualizations across all categories.

After that, we conducted a post hoc test to test pairwise comparisons (Table 5). The results show that there is a statistically significant difference in responses for category U3 between VIZB-VIZD, VIZB-VIZA, and VIZB-VIZC (Figure 2). Question U3 concerned the attractiveness of the color composition. Although the same color palette was used for all four visualizations, we can note that there is a statistically significant difference in VIZB-related responses (AS = 2.94). In addition, there is a statistically significant difference in responses for the TA category between VIZA-VIZB, VIZC-VIZB, VIZA-VIZD, and VIZC-VIZD (Figure 2). The number of taps (TA) gives us information about the type of interaction. It is evident from the results that there is a statistically significant difference in responses between different types of interactions. Thus, we can say that the drag interaction requires fewer taps.

The differences in answers are noticeable in the boxplot view of all visualizations according to the type of graph and answers for questions related to understanding and enjoyment (Figure 3). We can spot the dispersion of VIZB responses.

After analyzing the categories individually, we also examined the results at the group level. Based on the heat map (Table 6), in which we ranked the four visualizations, we recommend using VIZA and VIZC for a visualization that excels in all four categories (memorability, readability, understanding, and enjoyment). We give preference to VIZA over VIZC because it has a higher rank for more questions, including the crucial understanding category. VIZD takes third place as it is not ranked lowest in any question, while VIZB takes last place, with the lowest rank in as many as seven questions. Nonetheless, participants showed a moderate level of memorability for all visualizations, while readability, understanding, and enjoyment were high for all visualizations.

When talking about the engagement category, which was related to interaction, we ranked the two types of interaction we examined based on the results from the heat map (Table 7). The drag interaction was ranked highest in almost all questions, while the touch interaction had lower ratings.

The difference in answers is visible in the boxplot view of all visualizations by graph type and answers for questions related to engagement (Figure 4). The subjective assessment of A1, A2, and A3 indicates that the responses are equivalent, but we can observe the distinction between drag and touch interaction from the average time spent on the task and the number of taps.

## 4. Discussions

Based on the presented results, we answered and elaborated on each of the research questions. The answer to the first research question (Q1) is that there is no statistically significant difference in the evaluation of the data presented in the visualization between the four types of radial visualizations. While we cannot claim that there is a distinction in the subjective evaluation of the four types of radial visualization, we can still utilize the variation in responses by type of visualization and interaction to inform the design of radial information visualizations. Given that VIZC has the highest values in the enjoyment category, we can conclude that VIZC is visually the most attractive to respondents and thus answer the second research question (Q2). Additionally, the findings indicate that VIZA and VIZC are the most comprehensible to viewers, thereby addressing the third research question (Q3). As for interaction, based on the results, we can answer the fourth research question (Q4) and conclude that drag interaction is more suitable for radial visualizations. Based on the presented results, we recommend the following guidelines for designing interactive radial mobile information visualizations:Understanding: If the primary goal of the visualization is to enable users to make quantitative comparisons between data points, such as identifying relative magnitudes or trends, we recommend using a radial bar chart or a half-radial bar chart. These chart types provide a clear, intuitive display of the data that allows users to draw conclusions quickly and accurately.Visual Appeal: For optimal visual appeal, we recommend using a half-radial bar chart, which balances the benefits of the radial format with a more conventional presentation. Avoid using the polar area chart, as it was found to be the least visually appealing option in our study.Readability: If the display of precise numerical values is a key requirement, we recommend using a radial bar chart, as it provides an accurate and easily interpretable representation of the data.Effectiveness: While the donut chart was found to be satisfactory in our study, we recommend using a radial bar chart or half-radial bar chart for better effectiveness across multiple performance metrics, including accuracy, efficiency, and user satisfaction.Interaction: To optimize the interaction experience for radial visualizations in mobile applications, we suggest utilizing drag interaction, which was found to be more effective than touch interaction in our study.Comprehensive Performance: To achieve high performance across multiple evaluation criteria, including memorability, readability, understanding, and enjoyment, we recommend using a radial bar chart or a half-radial bar chart. These chart types offer a strong balance of accuracy, readability, and visual appeal.Decision-making: In cases of uncertainty in choosing between different radial visualizations, we recommend prioritizing the radial bar chart over the half-radial bar chart, as it had higher rankings in more questions, particularly in the crucial understanding category. The donut chart ranked third overall, as it was not ranked lowest in any question, while the polar area chart ranked last with the lowest rankings in as many as seven questions.

These guidelines are based on the empirical findings from the research presented in this paper and can serve as valuable recommendations for designers in designing effective radial mobile information visualizations.

## 5. Conclusions

In conclusion, our study provided a detailed analysis of four types of radial visualizations, namely radial bar charts, polar area charts, half-radial bar charts, and donut charts, along with two interactive techniques, drag and touch interactions. Our evaluation was conducted in the context of a fitness mobile application, and we assessed the perception of each design based on five main categories, namely memorability, readability, understanding, enjoyment, and engagement. Our results showed no statistically significant difference in responses by type of visualization or interaction, indicating that all four types are suitable for use within mobile activity tracking applications. However, our analysis also revealed the distinguishing characteristics of each visualization type in different categories. For example, the half-radial bar chart was found to be the most visually appealing and effective for understanding the data, while the radial bar chart was more suitable for readability. The polar area chart was perceived as the least appealing, while the donut chart was satisfactory in all categories but still not as effective as the radial and half-radial bar charts. Furthermore, we suggest utilizing drag interaction for mobile applications that require time changes. Our findings provide important insights into the use of radial visualizations in mobile applications, and we believe that our study will be useful for designers and researchers alike. Overall, our study’s contributions include providing design and methodological guidelines for similar research and expanding our understanding of the characteristics and effectiveness of radial visualizations in mobile applications.

### Limitations and Further Research

Unmoderated tests come with limitations, such as difficulty controlling the environment and conditions in which the subjects are located. Additionally, depending on the screen size of a mobile device, differences in perception may occur due to varying visualization sizes. The limitations of our research are sample size, display size, and type of visualization. For future research, we propose conducting experiments on a larger sample size to obtain more robust conclusions. Furthermore, testing other types of visualizations commonly used in mobile applications (e.g., bar graphs, line graphs, scatter graphs) and examining other types of interactions that were not included in this study should be considered. In addition, for more objective results, we recommend incorporating heat maps and eye tracking into the methodology. They can provide a more detailed understanding of users’ visual attention, how long they focus on specific elements, and the order in which they view them.

## Figures and Tables

**Figure 1 jimaging-09-00100-f001:**
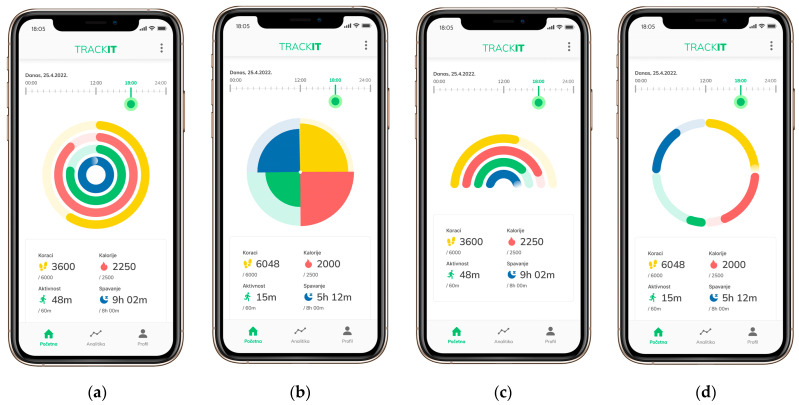
Visualizations designed for experimental research: (**a**) radial bar chart (VIZA), (**b**) polar area chart (VIZB), (**c**) half-radial bar chart (VIZC), and (**d**) donut chart (VIZD).

**Figure 2 jimaging-09-00100-f002:**
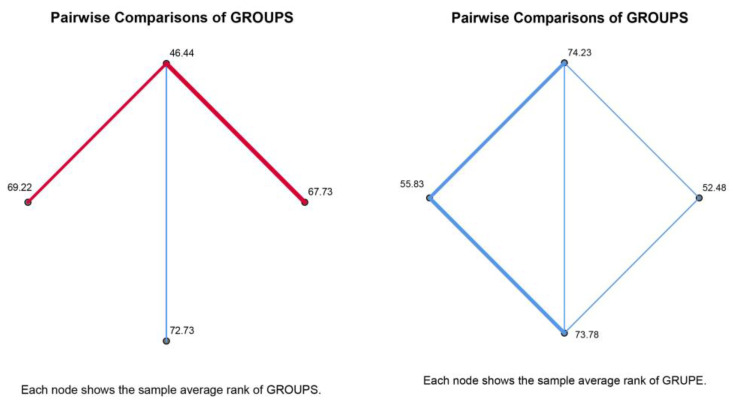
Pairwise comparison, (**a**) U3 and (**b**) TA (Adj. Sig., Bonferroni correction).

**Figure 3 jimaging-09-00100-f003:**
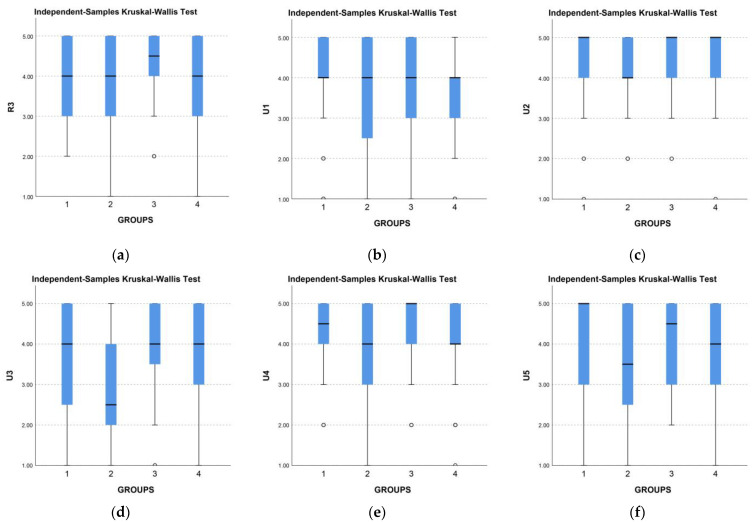
Comparison of results by types of visualizations (1-VIZA, 2-VIZB, 3-VIZC, 4-VIZD) for questions related to understanding, circles (○) represent far outliers, (**a**) R3, and enjoyment, (**b**) U1, (**c**) U2, (**d**) U3, (**e**) U4, (**f**) U5.

**Figure 4 jimaging-09-00100-f004:**
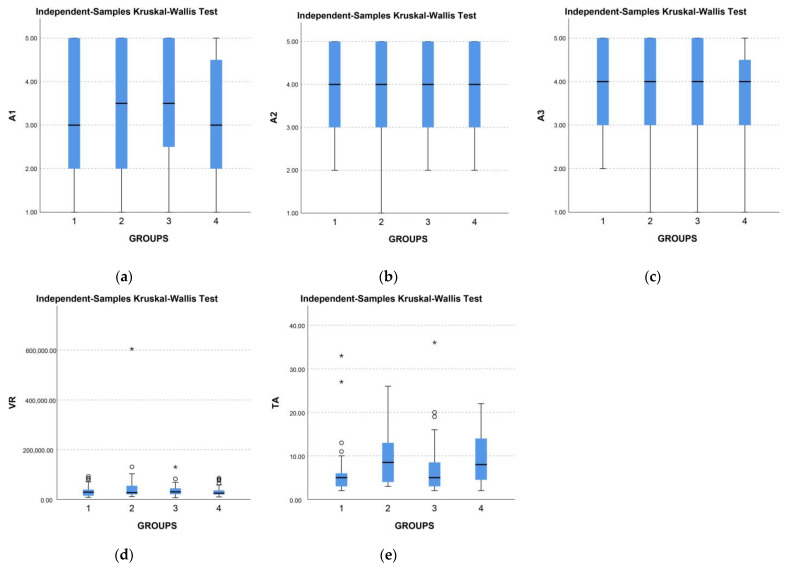
Comparison of results by types of visualizations (1-VIZA, 2-VIZB, 3-VIZC, 4-VIZD) for questions related to engagement, circles (○) represent far outliers, starred points (*) represent far out outliers, (**a**) A1, (**b**) A2, (**c**) A3, (**d**) VR, (**e**) TA.

**Table 1 jimaging-09-00100-t001:** Comparative analysis of data sets and visualization of 16 fitness applications, ind = individual.

Mobile App	Visualization Type	Data Type	Data Set (No. of Categories)	Colors	
VeryFit	radial bar chart	numerical	4	green, blue, red, purple	■ ■ ■ ■
Zeroner	radial bar chart	numerical	4	green, blue shade 1, blue shade 2, red	■ ■ ■ ■
Google Fit	radial bar chart	numerical	2	green, blue	■ ■
Health, diet and fitness—losing weight	radial bar chart	numerical	3	green, red, yellow	■ ■ ■
Vfit	radial bar chart		1	blue	■
CoolWear Fit	radial bar chart	numerical	1	-	
Fit Companion	radial bar chart	numerical	4	blue, red, purple, orange	■ ■ ■ ■
Samsung Health	radial bar chart	numerical	3	green, blue, pink	■ ■ ■
Walkingspree	radial bar chart (ind)	numerical	4 (ind)	green, blue, purple, orange	■ ■ ■ ■
Oplayer Smart Life	radial bar chart (ind)	numerical	6 (ind)	green, red, purple, yellow, orange, pink	■ ■ ■ ■ ■ ■
Fitbit	radial bar chart (ind)	numerical	5 (ind)	blue	■
LG Health	polar area chart	numerical	3	blue shade 1, blue shade 2, purple	■ ■ ■
OnePlus Health	polar area chart	numerical	4	green, blue, orange, pink	■ ■ ■ ■
ZeFit4	polar area chart	numerical	1	green, blue shade 1, blue shade 2	■ ■ ■
Joywear 2	half-radial bar chart	numerical	3	green, blue, orange	■ ■ ■
Letsfit	donut chart	numerical	3	green, blue, orange	■ ■ ■

**Table 2 jimaging-09-00100-t002:** Survey questions and their encodings: Likert scale, 1–5, 1—incomprehensible, 5—understandable (R3); Likert scale, 1–5, 1—strongly disagree, 5—strongly agree (U1, U2, U3, U4, U5, A1, A2, A3).

Code	Question	Answers
P1	Which of the four categories was the furthest from meeting the goal at noon?	Single select
P2	Which of the four data categories remained unchanged in the 12–18 h range?	Single select
C1	Which of the four data categories in the visualization met the goal?	Multi-select
C2	Which of the four categories in the visualization is furthest from meeting the goal?	Multi-select
R1	Which two categories in the visualization are approximately equally filled (close to goal fulfillment)?	Multi-select
R2	What is the difference between calories and activity in the visualization?	Single select
R3	Evaluate the degree of understanding of the displayed radial visualization.	Likert scale
U1	All elements of this prototype are aligned.	Likert scale
U2	The arrangement of colors on the radial visualization is pleasantly diverse.	Likert scale
U3	The composition of colors on the radial visualization is visually attractive.	Likert scale
U4	The prototype seems professionally designed.	Likert scale
U5	The radial visualization is visually appealing.	Likert scale
A1	Using a gesture to modify the display by hours is not difficult.	Likert scale
A2	Using a gesture to modify the display by the hour is intuitive.	Likert scale
A3	The gesture to modify the display by the hours is of expected behavior.	Likert scale

**Table 3 jimaging-09-00100-t003:** Comparison of all visualizations (VIZA, VIZB, VIZC, and VIZD)—the percentage of accuracy (P1, P2, C1, C2, R1, R2), degree of agreement with the statements (R3, U1, U2, U3, U4, U5, A1, A2, A3, Likert scale, 1–5), time on task (VR) and taps on task (TA), N = number of responses, % = percentage of correct answers, M = average values, CI = 95% confidence interval, SD = standard deviation, and GM = geometric mean.

Category	Code	Unit	VIZA	VIZB	VIZC	VIZD
		N	32	32	32	31
Memorability	P1	%	28%	50%	44%	42%
	P2	%	38%	22%	34%	23%
Readability	C1	%	91%	63%	63%	84%
	C2	%	94%	88%	88%	90%
Understanding	R1	%	81%	78%	50%	83%
	R2	%	59%	72%	38%	71%
	R3	M	3.97	3.63	4.19	3.74
		CI	(3.58, 4.35)	(3.15, 4.10)	(3.83, 4.55)	(3.29, 4.20)
		SD	1.03	1.31	1.00	1.24
Enjoyment	U1	M	4.06	3.63	3.97	3.71
		CI	(3.62, 4.51)	(3.09, 4.16)	(3.53, 4.40)	(3.32, 4.10)
		SD	1.20	1.48	1.20	1.07
	U2	M	4.29	4.16	4.53	4.35
		CI	(3.92, 4.66)	(3.85, 4.46)	(4.26, 4.81)	(4.01, 4.70)
		SD	1.01	0.85	0.76	0.95
	U3	M	3.81	2.94	4.00	3.84
		CI	(3.25, 4.30)	(2.41, 3.46)	(3.55, 4.45)	(3.39, 4.28)
		SD	1.42	1.46	1.24	1.21
	U4	M	4.25	3.84	4.41	4.10
		CI	(3.89, 4.56)	(3.42, 4.26)	(4.09, 4.72)	(3.69, 4.50)
		SD	0.92	1.17	0.87	1.11
	U5	M	4.00	3.47	4.13	3.71
		CI	(3.49, 4.45)	(2.98, 3.95)	(3.75, 4.50)	(3.24, 4.18)
		SD	1.30	1.34	1.04	1.27
Engagement	A1	M	3.28	3.38	3.63	3.19
		CI	(2.71, 3.81)	(2.87, 3.88)	(3.13, 4.12)	(2.72, 3.67)
		SD	1.49	1.41	1.39	1.30
	A2	M	3.94	3.81	3.84	3.84
		CI	(3.50, 4.31)	(3.42, 4.21)	(3.42, 4.26)	(3.47, 4.21)
		SD	1.11	1.09	1.17	1.00
	A3	M	3.91	3.59	3.97	3.68
		CI	(3.48, 4.26)	(3.14, 4.05)	(3.55, 4.38)	(3.29, 4.06)
		SD	1.06	1.27	1.15	1.05
	VR	M (s)	35	57	36	36
		GM (s)	28	34	29	28
		CI	(26, 44)	(19, 94)	(19, 94)	(27, 45)
	TA	M	6.6	9.5	7.3	9.6
		CI	(4.11, 9.11)	(7.23, 11.83)	(4.77, 9.86)	(7.36, 11.93)
		Interaction type	drag	touch	drag	touch

**Table 4 jimaging-09-00100-t004:** Non-parametric Kruskal–Wallis test, significance level 0.050, bold values represent statistically significant difference; H = Kruskal–Wallis H test; df = degree of freedom; *p* = *p*-value.

	R3	U1	U2	U3	U4	U5	A1	A2	A3	VR	TA
H	3.720	3.385	4.444	10.910	4.714	5.233	1.798	0.339	2.321	0.600	9.497
df	3	3	3	3	3	3	3	3	3	3	3
*p*	0.293	0.336	0.217	**0.012**	0.194	0.155	0.615	0.953	0.508	0.896	**0.023**

**Table 5 jimaging-09-00100-t005:** Pairwise comparison, bold values represent the pairs between which a statistically significant difference exists, *p* = *p*-value (Adj. Sig., Bonferroni correction).

	Sample 1-Sample 2	*p*		Sample 1-Sample 2	*p*
U3	VIZB-VIZD	**0.101**	TA	VIZA-VIZC	1.000
	VIZB-VIZA	**0.059**		VIZA-VIZB	**0.119**
	VIZB-VIZC	**0.017**		VIZA-VIZD	**0.109**
	VIZD-VIZA	1.000		VIZC-VIZB	**0.296**
	VIZD-VIZC	1.000		VIZC-VIZD	0.275
	VIZA-VIZC	1.000		VIZB-VIZD	1.000

**Table 6 jimaging-09-00100-t006:** Rating by category (memorability, readability, understanding, and enjoyment); ■ = first in the ranking (highest percentage of accuracy or highest score of agreement on the Likert scale); ■ = second in the ranking; ■ = third in the ranking; □ = fourth in the ranking (minimum percentage of accuracy or minimum score of agreement on the Likert scale); % = percentage of correct answers; M = average values.

Category	Code	Unit	VIZA	VIZB	VIZC	VIZD
Memorability	P1	%	28%	50%	44%	42%
	P2	%	38%	22%	34%	23%
Readability	C1	%	91%	63%	63%	84%
	C2	%	94%	88%	88%	90%
Understanding	R1	%	81%	78%	50%	83%
	R2	%	59%	72%	38%	71%
	R3	M	3.97	3.63	4.19	3.74
Enjoyment	U1	M	4.06	3.63	3.97	3.71
	U2	M	4.29	4.16	4.53	4.35
	U3	M	3.81	2.94	4.00	3.84
	U4	M	4.25	3.84	4.41	4.10
	U5	M	4.00	3.47	4.13	3.71

**Table 7 jimaging-09-00100-t007:** Rating by category (engagement), ■ = first in the ranking (highest percentage of accuracy or highest score of agreement on the Likert scale), ■ = second in the ranking, ■ = third in the ranking, □ = fourth in the ranking (minimum percentage of accuracy or minimum score of agreement on the Likert scale), % = percentage of correct answers, M = average values.

Category	Code	Unit	Drag (VIZA)	Touch (VIZB)	Drag (VIZC)	Touch (VIZD)
Engagement	A1	M	3.28	3.38	3.63	3.19
	A2	M	3.94	3.81	3.84	3.84
	A3	M	3.91	3.59	3.97	3.68
	VR	GM (s)	28	34	29	28
	TA	M	6.6	9.5	7.3	9.6

## Data Availability

The data presented in this study are available on request from the corresponding author.

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
