# Peer review of "Assessing the Design of Interactive Radial Data Visualizations for Mobile Devices"

_2313-433X, 2023, doi:10.3390/jimaging9050100_

Round 1
Reviewer 1 Report
1) The article is basically interesting, but in my opinion, paragraph after paragraph, its engineering character is "blurred". In my opinion, the beginning is compact and concrete. Meanwhile, the design guidelines and summary are relatively general. Please comment.
2) In my opinion, the abstract is too general. According to J.Imaging: The abstract should be a single paragraph and should follow the style of structured abstracts, but without headings: 1) Background: Place the question addressed in a broad context and highlight the purpose of the study; 2) Methods: Describe briefly the main methods or treatments applied. Include any relevant preregistration numbers, and species and strains of any animals used. 3) Results: Summarize the article's main findings; and 4) Conclusion: Indicate the main conclusions or interpretations. The abstract should be an objective representation of the article: it must not contain results which are not presented and substantiated in the main text and should not exaggerate the main conclusions. Please comment.
3) What is the relationship between the charts in the TRACKIT application and dashboard systems? Dashboard is a visualization tool that allows you to present key information from various sources in one place. It is usually an interactive board that allows you to easily and quickly track selected indicators and metrics. Depending on the purpose and needs of the user, dashboards can contain various types of charts, tables, maps, as well as interactive elements such as sliders or filters that allow you to adjust the view to specific needs. Please comment.
4) Research results seem relative. How much do research results depend on the subjective characteristics of the respondents? How much do test results depend on the size of the display? Please comment.
5) Is the research sample (number of users) sufficient? In my opinion, a larger research sample is needed for statistical inference – the minimum random sample size estimated at 386 questionnaires (for the margin of error of ±5% and confidence level p = 0.95). Please comment.
6) I suggest improving section "5. Conclusions”. In my opinion, as it stands, the content in this section is too general. Moreover, at the end of the section, the authors "suddenly" mention heat maps and eye tracking. Consequently, section 5 should be split into: 5. Conclusions and 5.1. Limitation and further research. Please comment.
Reviewer 2 Report
This paper investigates the design of interactive visualizations on mobile devices and new evaluation methods for interactive radial visualizations. An empirical study to assess the perception of four types of circular visualizations on mobile devices has been carried out. The authors found no statistically significant difference in the different visualizations.
The paper is interesting, technically solid and perfectly fits the scope of the journal. I have just some minor notes:
- The first Figure is not readable and has no caption
- The text in Figures 2, 3 is too blurry
A proofread is needed to correct errors and typos
Round 2
Reviewer 1 Report
1) The article is basically interesting, but in my opinion, paragraph after paragraph, its engineering character is "blurred". In my opinion, the beginning is compact and concrete. Meanwhile, the design guidelines and summary are relatively general. Please comment.
Reviewer comment: The authors' corrections and commentary are appropriate.
2) In my opinion, the abstract is too general. According to J.Imaging: The abstract should be a single paragraph and should follow the style of structured abstracts, but without headings: 1) Background: Place the question addressed in a broad context and highlight the purpose of the study; 2) Methods: Describe briefly the main methods or treatments applied. Include any relevant preregistration numbers, and species and strains of any animals used. 3) Results: Summarize the article's main findings; and 4) Conclusion: Indicate the main conclusions or interpretations. The abstract should be an objective representation of the article: it must not contain results which are not presented and substantiated in the main text and should not exaggerate the main conclusions. Please comment.
Reviewer comment: The authors' corrections and commentary are appropriate.
3) What is the relationship between the charts in the TRACKIT application and dashboard systems? Dashboard is a visualization tool that allows you to present key information from various sources in one place. It is usually an interactive board that allows you to easily and quickly track selected indicators and metrics. Depending on the purpose and needs of the user, dashboards can contain various types of charts, tables, maps, as well as interactive elements such as sliders or filters that allow you to adjust the view to specific needs. Please comment.
Reviewer comment: The authors' corrections and commentary are appropriate.
4) Research results seem relative. How much do research results depend on the subjective characteristics of the respondents? How much do test results depend on the size of the display? Please comment.
Reviewer comment: The authors' corrections and commentary are appropriate.
5) Is the research sample (number of users) sufficient? In my opinion, a larger research sample is needed for statistical inference – the minimum random sample size estimated at 386 questionnaires (for the margin of error of ±5% and confidence level p = 0.95). Please comment.
Reviewer comment: The authors' corrections and commentary are appropriate.
6) I suggest improving section "5. Conclusions”. In my opinion, as it stands, the content in this section is too general. Moreover, at the end of the section, the authors "suddenly" mention heat maps and eye tracking. Consequently, section 5 should be split into: 5. Conclusions and 5.1. Limitation and further research. Please comment.
Reviewer comment: The authors' corrections and commentary are appropriate.